# Assessment of the effectiveness of host depletion techniques for profiling fish skin microbiomes and metagenomic analysis

Ashley G. Bell,[1,2] Jo Cable,[3] Ben Temperton,[1] Charles R. Tyler[1,2]

**ABSTRACT**  Microbiomes on fish mucosal surfaces play crucial roles in nutrient absorption, immune priming, and defense, and disruptions in these microbial communities can lead to adverse health outcomes, including disease. Studying fish microbiomes relies on sequencing microbiota within mucosal-rich samples; however, nucleic acid extraction from these samples is composed predominantly of host DNA, making subsequent bioinformatic processes difficult. Host depletion techniques address this issue by either selectively degrading host DNA before sequencing or retaining bacterial DNA post-extraction. However, their application to fish mucosal samples has been largely unexplored. Here, we assessed the efficacy of various host depletion techniques on fish skin mucosal swabs via either selectively removing CpG-methylated (predominantly eukaryotic) DNA or selectively lysing eukaryotic cells before DNA extraction. Surprisingly, none of the existing methods we assessed effectively reduced host DNA to be practically useful. Furthermore, some methods introduced a bias toward certain bacterial taxa, including the Bacilli class and the Proteobacteria phylum. Our findings illustrate that the currently available host depletion techniques are largely ineffective for reducing host DNA in fish mucosal samples. This poses a major limitation for developing an understanding of the functional composition of fish mucosal microbiomes, as enriching microbiota (and excluding host DNA) is fundamental for cost-effective metagenomic studies and facilitating more accurate analyses of the microbiota metabolome and proteome.

**IMPORTANCE**  Microbial communities on fish mucosal surfaces are vital for immune function and disease resistance. However, sequencing these communities is hindered by the dominance of host DNA in mucosal samples, which can exceed 99% of total nucleic acids. While host depletion techniques are routinely used in human and mammalian systems to enrich microbial DNA, their efficacy on fish samples remains uncharacterized. In this study, we assessed multiple commercial and published host depletion methods on fish skin microbiomes. None significantly reduced host DNA to levels suitable for high-quality metagenomic sequencing, and some introduced taxonomic bias. We suggest methodological reasons, including differences in fish cell structure and mucus composition compared to mammalian systems, that may explain these shortcomings. Based on our findings, we propose protocol modifications and highlight key areas for improvement. This work identifies critical limitations and offers a foundation for developing optimized host depletion strategies tailored to fish mucosal microbiome research.

**KEYWORDS**  fish skin, microbiome, host DNA depletion, metagenomics, metagenome-assembled genomes (MAGs), microbial DNA enrichment, microbial community profiling, *Oncorhynchus mykiss*

Address correspondence to Ashley G. Bell, agb214@exeter.ac.uk, or Charles R. Tyler, C.R.Tyler@exeter.ac.uk.

The authors declare no conflict of interest.

See the funding table on p. 18.

10.1128/spectrum.01838-25   1

Microbiomes are important for host development, regulation of host health, and protection from infectious diseases (1–3). Microbiomes in fish mucosal surfaces play key roles in disease resilience and adapting to physicochemical changes in the environment (4, 5). Disruptions to these microbiomes can result in the depletion of taxa involved in nutrient absorption (6), immune priming (7, 8), and altered behavior (9). There is now considerable research into probiotics (10) and alternative diets (11, 12) for optimizing microbiomes. Methods for studying microbiomes include the application of next-generation sequencing, facilitating high-throughput analysis of complex communities and allowing researchers to capture the widest diversity, not revealed by culturing methods (2, 13).

Amplification and classification of 16S and 18S rRNA is a widely used tool to study microbiomes, allowing for the identification of microorganisms captured within a metagenomic sample. This technique, however, relies on the amplification of target sequences, which can be problematic when amplifying DNA from samples dominated by large amounts of host DNA (14). Successful amplification of target (microbiota) DNA from such samples requires high starting concentrations of DNA, but this can also include higher concentrations of PCR inhibitors and non-specific DNA (15, 16). This, in turn, can reduce PCR amplification efficiency and result in amplification of unintended DNA due to similarities (e.g., mitochondrial DNA when targeting 16S rRNA) or necessitate the use of more expensive, higher fidelity polymerases (17). Amplicon sequencing can also be biased depending on primer selection (18) and can furthermore lead to base substitutions and chimeras (19–21). Importantly, amplicon sequencing does not allow for the study of whole genomes for genomic and functional studies. For this purpose, shotgun metagenomics is employed, which seeks to sequence a subset of all genetic material within a sample.

Assembly of shotgun metagenomic data into long contiguous sequences improves strain-level divergence determination due to higher accuracy from the longer read lengths (22, 23). Compared with amplicon sequencing, shotgun metagenomics requires a higher initial DNA concentration and more sequencing data to achieve the same sequencing depth, which can be a disadvantage for low-yield samples. This becomes particularly costly for samples with high host DNA contamination, such as saliva and mucosal swabs, as untreated mucosal samples often contain over 90% of host DNA sequences (24–26). Although some studies have reported successful metagenomic recovery from fish skin swabs, they note low microbial sequencing coverage as a factor, in turn, impacting metagenomic studies (27).

Host depletion methods that reduce host DNA without degrading bacterial DNA offer significant advantages as they can increase microbial sequencing coverage, reduce financial costs, and allow for more contiguous assemblies due to higher microbial sequencing depth (24, 28, 29). This, in turn, improves the quality of metagenomically assembled genomes (MAGs) and facilitates the detection of rare taxa. Various commercial kits and studies have been developed that successfully deplete eukaryotic (i.e., host) DNA sequences, thereby increasing the proportion of microbial DNA recovered within metagenomic samples (24, 25, 29) and have been applied to fish samples (30). DNA from fish skin swabs can be comprised of up to 99% fish skin DNA (31), and studies on skin microbiota would benefit greatly from such host depletion methods. Fish skin studies have mostly been limited to amplicon sequencing methods, which are not limited by the high host DNA content. However, depletion of non-target sequences like mitochondrial DNA would lower off-target amplification and also reduce total concentrations of PCR inhibitors. This, in turn, would decrease required PCR DNA concentrations and, in turn, increase the relative amount of microbial DNA in the sample (26, 32). Popular DNA depletion kits and methods include the NEBNext Microbiome DNA Enrichment Kit (E2612L) from New England BioLabs, Molzym's Molysis kits (D-321-050), Zymo's HostZERO microbial DNA kit (D4310), and the selective lysis and degradation of eukaryotic cell DNA described by Hunter et al. (33). In this study, we assess these three commercial kits (The *NEBNext Microbiome DNA Enrichment Kit*, the "*Molysis*" method, and

the "*Zymo*" method—a shortened version of the HostZERO Microbial DNA Kit [D4310]) as well as the *Hypotonic lysis method* by Hunter et al. (33) with modifications by Marotz et al. (24), and Nelson et al. (25) and an alternative to the Hypotonic Lysis method employing phosphate-buffered saline (PBS) instead of deionized water. These latter studies detail the effective depletion of human DNA from saliva and sputum samples, which share similar physical properties to fish mucosal samples.

In this study, the above five methods (detailed in the Materials and Methods section) were used to determine the effectiveness of different host depletion techniques on fish skin mucosal samples. For each sample, we used qPCR to assess the log-fold change of bacterial to host template DNA. Specifically, we compared the quantification cycle (Cq) amplification of 16S and 18S rRNA using custom-designed trout-specific 18S primers and established bacterial 16S primers (34) to determine the relative depletion of 18S rRNA compared to 16S amplification. Amplicon analysis of both 16S and 18S rRNA genes was then used to determine the microbial community composition from each host depletion method compared with a control without host depletion techniques applied. Using this amplicon analysis, we also assessed whether taxa were differentially abundant compared to the control group, indicating which microbial groups were enriched or depleted by the depletion methods. In the last part of this analysis, we employed shotgun metagenomics to determine the percentage of nucleotides identified as host (fish) DNA. This approach assessed the effectiveness of the host depletion methods in degrading host DNA and their ability to facilitate the construction of high-quality MAGs.

## MATERIALS AND METHODS

### Fish husbandry

Juvenile rainbow trout (*Oncorhynchus mykiss*) (15–25 cm in length, 150–200 g weight) were obtained from a local commercial fish farm and housed at the University of Exeter Aquatic Resource Center. Trout were housed in 250 L gray opaque plastic tanks with clear lids that received dechlorinated freshwater on a recirculating system. Tank water was aerated, maintained at 13°C ±1°C, and monitored for pH, temperature, conductivity, dissolved oxygen, ammonia, nitrate, nitrite, and alkalinity weekly, which were within the guidelines set by the U.S. EPA (35). The photoperiod was set to a 12-h light/dark cycle (0800:2000) with a 30-min artificial dawn/dusk transition.

### Fish skin swab sample collection

For the host DNA depletion studies, the laboratory-held rainbow trout were sampled on three occasions (9th September 2022, 10th November 2022, and 1st December 2022). Skin mucosal DNA material was collected from individual fish onto individual sterile 47 mm filter paper discs (Whatman; 7141114); pilot studies had established that cotton swabs did not yield sufficient DNA from skin samples when this was depleted for high-throughput sequencing. In the collection process, the trout were first netted from their tanks, and a filter paper disc, held with sterilized forceps, was rubbed along the trout's lateral line from the point on the body of the dorsal fin in an anterior-to-posterior direction. This was repeated three times with each filter paper disc. Each fish in the laboratory-maintained population was sampled once only over the three collection times, yielding one disc per fish. A total of 87 fish were sampled, producing 87 independent samples. These were distributed as evenly as possible across the different host depletion methods and were processed immediately according to the DNA depletion methods (Table 1).

### DNA extraction and quantification

All DNA extractions were performed and completed the day after sample collection. For samples undergoing the CpG-methylated method (host depletion after DNA extraction), filters were stored in 570 µL lysis buffer (30 mM Tris, 30 mM EDTA, pH 8) at −80°C for at

**TABLE 1** DNA yield, purity, and fragment size obtained for the DNA extractions following different host depletion methods

| Method | 9th September 2022, N | 10th November 2022, N | 1st December 2022, N | DNA quantity (ng/µL) | | | 260/280 ratio | 260/230 ratio |
|---|---|---|---|---|---|---|---|---|
| | | | | %5 | Median | %95 | | |
| Standard DNA extraction | 3 | 8 | 6 | 111.40 | 291.00 | 463.40 | 2.00 | 1.98 |
| CpG-methylated | 3 | 8 | 6 | 6.53 | 17.80 | 60.63 | 1.97 | 1.69 |
| Hypotonic lysis | 2 | 6 | 7 | 23.96 | 57.80 | 134.90 | 2.02 | 1.73 |
| PBS | 0 | 0 | 8 | 20.60 | 68.75 | 106.37 | 1.95 | 1.74 |
| Molysis | 2 | 7 | 5 | 0.70 | 3.08 | 91.73 | 1.92 | 0.839 |
| Zymo | 2 | 7 | 7 | 2.25 | 12.60 | 47.82 | 1.90 | 1.68 |

least 1 h before DNA extraction to facilitate freeze-thaw cell lysis. For methods requiring depletion before DNA extraction, this was performed prior to storage in lysis buffer as detailed above and at −80°C. DNA extraction was performed using an in-house CTAB/EDTA/chloroform method adapted from Chaput (36) and Lever et al. (37) (full protocol available at https://dx.doi.org/10.17504/protocols.io.bw8gphtw). Extracted DNA was resuspended in 50 µL Qiagen Elution Buffer (10 mM Tris-Cl, pH 8.5), equilibrated for 5 min at room temperature, quantified, and stored at −20°C. DNA concentration was quantified using 2 µL of extracted DNA with a ThermoFisher Qubit dsDNA Broad Range Assay Kit, following the manufacturer's protocol. To determine DNA purity, 1 µL of extracted DNA was used for spectrophotometric analysis using 260/280 and 260/230 absorbance ratios in a NanoDrop 1000 (Thermo Scientific; ND-1000), following the manufacturer's protocol.

## DNA depletion

Here, we first describe the five host DNA depletion methods trialed. The NEBNext Microbiome DNA Enrichment Kit removes CpG-methylated DNA, which is common in eukaryotes but rare in bacteria (38) and is the basis for the "CpG-methylated" method. CpG-methylated DNA is bound to magnetic beads, and the microbial DNA is left free in the supernatant. Using this method, host DNA depletion has been shown to be effective in sequencing malaria by depleting human DNA from "clinical isolates" (28). It has also been successfully applied to enrich microbes present within whole homogenized black molly (*Poecilia sphenops*) samples (30).

The "Molysis" method is the basis for the MolYsis Basic5 kit from Molzym and uses a proprietary chaotropic buffer to selectively lyse eukaryotic cells and an endonuclease to degrade released DNA before the sample undergoes DNA extraction. This has been shown to be effective at degrading host DNA in both bovine and human milk samples (39), saliva (24), and sonicated fluids from prosthetic joint infections (40).

The "Zymo" method is a shortened version of the HostZERO Microbial DNA Kit (D4310) and uses a Host Depletion Solution (D4310-1-20) to selectively lyse eukaryotic cells and a Microbial Selection Buffer (D4310-2-5) and Enzyme (D4310-3-50) to degrade DNA and RNA before DNA extraction. The Zymo method was shown to be effective in the removal of human DNA in diabetic foot infections (29).

The "Hypotonic Lysis" method was designed and refined by Hunter et al. (33) and Marotz et al. (24) and targets the inability of eukaryotic cells to regulate osmosis in deionized water. Extracellular DNA and DNA from lysed eukaryotic cells are degraded by an endonuclease or sequestered by propidium monoazide before DNA extraction. This has been shown to be effective at depleting human DNA in saliva samples (24, 33), human cerebrospinal fluid samples (41), and human sputum samples (25), but not in human milk samples (14).

We also evaluated an alternative to the Hypotonic Lysis method employing PBS instead of deionized water. This aimed to understand the impact of washing (loss) of host cells when employing host depletion solutions. Host depletion techniques may lead to a decrease in host DNA, which is washed off instead of being degraded. PBS was selected as this is a salt-based solution with no degrading effects on microbial or host cells (42).

All DNA depletion methods were performed immediately after sample collection except for the CpG-methylated method, which was performed immediately after DNA extraction. For the CpG-methylated method, DNA was resuspended in an elution buffer and processed with NEBNext Microbiome DNA Enrichment Kit (NEB E2612S) according to the manufacturer's instructions (43). For the Hypotonic Lysis method, the DNA depletion method was modified from Hunter et al. (33), Marotz et al. (24) and Nelson et al. (25) to accommodate filters. Before DNA extraction, filters were suspended in 500 µL of deionized ultrapure autoclaved water (NFW) and incubated on a thermal mixer for 1 h at 24°C and 600 rpm. Benzonase buffer (20 mM Tris-HCl pH 8, 1 mM MgCl$_2$ final) and 15.625U benzonase were added, and the samples were incubated on a thermal mixer at 37°C for 2 h at 600 rpm. The benzonase reaction was quenched by adding EDTA (5 mM final) and NaCl (150 mM final), and the resulting solution was washed twice by centrifugation at 8,000 × $g$ for 10 min at 4°C. The pellet was then washed and resuspended in 500 µL NFW, followed by final centrifugation at 8,000 × $g$ for 10 min at 4°C to pelletize. The NFW supernatant was removed and discarded, and the pellet was resuspended in 570 µL lysis buffer and stored at −80°C overnight. The following day, the DNA from the sample was extracted as described in the DNA extraction protocol above. The PBS method was identical to the Hypotonic Lysis method, except that in all steps, NFW was replaced with PBS. For the Molysis method, before DNA extraction, swabs were processed with Molzym's Molysis Basic kit (Molzym; D-300-008) stopping after the manufacturer's protocol final washing of the bacterial pellet (step 5) when the pellet was then suspended in 570 µL lysis buffer and stored at −80°C overnight before DNA extraction the following day as described in the above DNA extraction protocol. For the Zymo method, before DNA extraction, swabs are incubated in Host Depletion Solution (D4310-1-20), stopping after the manufacturer's protocol step 8 part (II) of the Host DNA depletion instruction section, when samples were stored in 570 µL lysis buffer at −80°C overnight and the DNA extracted the next day as described in the above DNA extraction protocol.

## Quantitative PCR

All qPCRs on samples for the different host depletion methods were performed on the same day. To assess the relative abundance of bacterial DNA compared to fish DNA, several 16S and 18S rRNA gene primers were tested for their efficacy in amplifying fish skin swab DNA. Primers targeting the 16S rRNA gene from Nadkarni et al. (34) (5′-TCCT ACGGGAGGCAGCAGT-3′ and 5′-GGACTACCAGGGTATCTAATCCTGTT-3′) were chosen due to their superior performance and consistency. qPCRs were performed in 15 µL volumes containing 7.5 µL of BioRad iTaq Universal SYBR green supermix (BioRad; 1725120), 5.75 µL of nuclease-free water (NFW), 0.25 mM each of forward and reverse primer, and 1 µL of DNA extract. The reaction was run on a BioRad C1000 CFX-96 thermal cycler with the following conditions: initial denaturation at 95°C for 10 min, followed by 40 cycles of 95°C for 15 s and 59.5°C for 30 s, as described in Nadkarni et al. (34). Novel primers targeting rainbow trout 18S rRNA were designed in-house (5′- GCTCGTAGTTGGATCTCGGG-3′ and 5′-GTTAAGAGCATCGAGGGGGC-3′) targeting positions 643–662 (forward) and 723–742 (reverse) with a resulting amplicon size of 100 base pairs. The reaction concentrations were identical to those used for 16S gene quantification but with different cycling conditions that included an initial denaturation step of 95°C for 3 min, followed by 40 cycles of 95°C for 10 s and 61.4°C for 30 s. The optimal annealing temperature of 61.4°C was determined through a gradient qPCR performed on randomly selected trout skin microbiome samples from this experiment. Negative controls (blank filters and no template controls) and positive controls (ZymoBIOMICS Microbial Community DNA Standard [Zymo; D6305]) were included in each qPCR assay. Melt curves were analyzed for each well to confirm the amplification of a single product. qPCR standard curves were performed for both amplicon products. A standard curve for 16S rRNA was performed on an *E. coli* DNA extraction at 63.5, 12.68, 6.34, 1.268, 0.634, 0.0634, and 0.00634 ng/µL. This was repeated for a trout fin clip DNA extraction at 200, 100, 50, 10, 1, and 0.1 ng/µL.

All standard curves were performed in duplicate, and efficacy values were confirmed to be close to 100%. 16S–18S rRNA expression fold change was calculated by the ΔCq method using the following formula $\Delta Ct = 2^{-(Cq_{16S} - Cq_{18S})}$. Fold changes were normalized by dividing by the median Standard DNA extraction ΔCt and plotted using a log10 transformation

## 16S rRNA V4 and 18S rRNA V9 amplicon library preparations and sequencing

All samples underwent preparation for amplicon sequencing using Earth Microbiome Project (EMP) primers targeting the 16S rRNA V4 region and 18S rRNA V9 region (44–48). Both 16S and 18S rRNA amplification were performed alongside controls, including positive controls, negative controls (filters, swabs), and no-template controls for each 96-well plate. Generation of 18S rRNA V9 amplicons employed a one-step PCR indexing approach and custom primers from the EMP (49). The V4 region primer pairs (515F–806R) of 16S rRNA were constructed (Integrated DNA Technologies), ensuring each well within a plate was dual indexed to reduce index hopping using updated 16S primers, barcodes, and linkers from the EMP (44–48) (Table S1). After 16S V4 amplification, an Illumina barcode was ligated to equimolar pooled amplicons from the same plate to allow for the use of multiple plates with the same indexes, an approach used in Guenay-Greunke et al. (50). All PCRs were performed in duplicate, each with a total reaction volume of 25 µL and consisting of 12.5 µL of Platinum SuperFi II Master Mix DNA Polymerase (Thermo-Fisher; 12368010) for 16S amplification or NEBNext High-Fidelity 2× PCR Master Mix (New England BioLabs; M0541) for 18S amplification, 9 µL of deionized water, 1.25 µL of forward and reverse primers each (0.5 µM final reaction concentration), and 1 µL of DNA extract. 16S V4 PCR conditions consisted of an initial denaturing step of 98°C 30 s, 30 cycles of denaturing at 98°C for 10 s, annealing at 61°C for 20 s, and extension at 72°C for 30 s, with a final extension step of 72°C for 2 min. 18S amplification followed identical PCR conditions except for an annealing temperature of 60°C. After amplification, duplicate PCRs were combined, and amplicons were checked for size using standard gel electrophoresis (16S V4 ~319 bps; 18S V9 ~260 bps), to ensure there were no spurious bands. Failed PCRs were repeated. PCR products were cleaned using a magnetic bead clean-up protocol outlined in reference (51) and equimolar pooled at 50 ng/µL. Amplicon concentration was determined using the QuantiFluor dsDNA System (E2670, Promega) in 96-well plate format. Pooled plates were gel-purified of non-target DNA, residual primer dimers, and other molecular matter using a Qiagen MinElute Gel Extraction Kit (28604) as outlined by the manufacturer's instructions. Gel-purified pools were confirmed to have amplicons of expected size and concentration using 1 µL of pooled DNA on the Agilent D1000 ScreenTape System and 2 µL Qubit Broad Range dsDNA Quantification Assay Kits (Invitrogen; Q32850), following the manufacturer's recommended protocol before submission to the University of Exeter Sequencing Service for Illumina barcoding (16S V4 only) and sequencing. 16S V4 barcoding consisted of PCR-free ligation of Illumina adaptors (plate barcoding) and amplicon sequencing on the NovaSeq SP 2 × 250 paired ends (500 cycles). 18S V9 amplicons already contained Illumina barcodes due to one-step PCR amplification, and after quantification, they were sequenced separately using MiSeq v2 2 × 150 paired ends (300 cycles). 50,000 reads per sample were targeted for all amplicons from fish skin swab samples.

## 16S amplicon sequence data processing

The data for this study have been deposited in the European Nucleotide Archive at EMBL-EBI under accession number PRJEB82663. All bioinformatic processes are available at https://github.com/ash-bell/host_depletion. After sequencing, 16S amplicon sequences were de-multiplexed using Illumina barcodes (to plate level) by the Exeter Sequencing Center. The cutadapt algorithm v4.5 (52) was then used to demultiplex at the sample (well) level, keeping only reads with both dual indices (Table S1) and allowing for a maximum of two mismatches on dual indices for sample identification. Any nucleotides with a quality score of Q2 or lower, containing "N"s as base pairs, or

that were identified as Phix sequences were removed. Forward reads were truncated at position 216 and reverse reads at 217 based upon manual inspection of read quality profiles in R via DADA2 v1.28.0 (53). As the NovaSeq error correction model has not been implemented in DADA2, four different error rates were learned using a minimum of 1 billion bases for both the forward and reverse reads. The loessErrfun_mod4, which alters the loess function and enforces monotonicity manually, was selected as the best model for correcting both the forward and reverse reads based on nucleotide error frequency versus consensus quality scores (54). Unique reads were dereplicated, and forward and reverse reads were merged into ASVs, producing an ASV count table. Chimeric sequences were removed, and only ASVs between 252 and 254 base pairs were retained. PCR-free ligation of Illumina adaptors resulted in reads occurring both in the forward and reverse complement orientations due to the non-specific binding of Illumina sequencing adaptors to either the forward or reverse orientation of reads. Around 50% of reads were in the reverse complement orientation and were oriented to the forward orientation before taxonomy was assigned using the SILVA database (55) based on the nr99_v138.1_train_set to genus level. A phylogenetic tree was constructed from ASVs using IQ-TREE v2.2.5 (56) algorithm with an extended model selection followed by tree inference.

## 16S amplicon sequence data analysis

All statistical analyses were performed in R v4.3.2 (57) with data manipulation performed using tidyverse v2.0.0 (58). Figures were constructed using ggplot2 v3.4.4 (59), ggsignif v0.6.4 (60), microViz 0.11.0 (61), ggh4x v0.2.7 (62), patchwork v1.2.0 (63), and ggpubr v0.6.0 (64). All linear model statistical tests were performed using lmerTest v3.1-3 (65) with the treatment group as the fixed effect and the sampling date as the random effect. The normality of linear model residuals was confirmed using a Q-Q plot and heteroskedasticity by plotting fitted values against the residuals. All statistical tests with multiple re-testing were corrected for multiple hypothesis testing using the Benjamini and Hochberg method (66). Amplicon data were manipulated using the phyloseq v1.46.0 (67), and phylogenetic trees were rooted using the longest branch using the ape v5.7-1 (68) to ensure reproducibility. Positive controls (ZymoBIOMICS Microbial Community DNA Standard) were compared with manufacturers' expected microbial community composition to ensure even DNA extraction (Fig. S1). Decontam v1.22.0 (69) was used to remove presumed contaminating ASVs identified either through the frequency or prevalence method at 0.1 and 0.25 thresholds, respectively. 16S ASVs identified at a family level as Mitochondria, order level as Chloroplast or domain level as Eukaryota were discarded. Alpha diversity was first rarified to the smallest sample (2,845 ASVs) with Shannon and Chao1 diversity measured using phyloseq and Faith's phylogenetic diversity using the picante v1.8.2 (70). Beta-diversity was calculated using microViz for a Bray-Curtis (vegan v2.6.4) (71), Weighted UniFrac (GUniFrac v1.8) (72, 73), Jaccard, or Aitchison dissimilarity matrix. Bray-Curtis dissimilarity matrices were calculated on a compositional transformed ASV table aggregated at a genus level, Weighted UniFrac on an ASV table with no transformation, Aitchison on an ASV table aggregated to a genus level, and Jaccard on a binary (presence absence) ASV table aggregated to a genus level. Ordination was performed using NMDS on a Bray-Curtis dissimilarity matrix. Beta-diversity statistics were calculated using a PERMANOVA with the adonis2 function from the vegan and post hoc style test performed using a pairwise.adonis2 v0.4.0 (74) wrapper script. Beta-dispersions of treatment groups calculated from a Bray-Curtis dissimilarity matrix were determined using microViz. Differential abundance of taxa was calculated using a linear model and corrected for multiple hypothesis testing as described above using microViz on compositional data. Only bacterial phylum, order, or class with a minimum total abundance of 1% across all samples was included in the differential abundance analysis.

**TABLE 2** Different effectiveness of host depletion methods as measured by qPCR 10-fold change against a control DNA extraction method using a linear model adjusted for multiple hypothesis testing[a]

| Comparison | | Estimate (10-fold difference from control) | Std. error | t-Value | P adjusted value | Significance |
|---|---|---|---|---|---|---|
| | CpG-methylated | 2.577 | 0.455 | 5.66 | <0.001 | *** |
| | Hypotonic Lysis | 1.857 | 0.469 | 3.96 | <0.001 | *** |
| Standard DNA extraction | Molysis | 2.789 | 0.478 | 5.84 | <0.001 | *** |
| | PBS | 1.428 | 0.563 | 2.54 | 0.014 | * |
| | Zymo | 2.782 | 0.813 | 3.42 | 0.002 | *** |

[a]$*P < 0.05$, $***P < 0.001$. Formula "log10(Normalized.Relative.Expression) − Treatment.group + (1|Sampling.date)".

## Shotgun metagenomic library preparations, sequencing, and data processing

Metagenomic samples were sent to the Exeter sequencing center for PCR-free genomic fragment library preparation and sequenced on the NovaSeq SP 2 × 250 paired ends (500 cycles) with a target of 10 million reads per sample. All DNA samples passed the Exeter Sequencing Center's quality control, which included assessment of high molecular weight DNA by agarose gel electrophoresis or Agilent D1000 ScreenTape Analysis, prior to sequencing. Metagenomes were processed applying quality control and error correction with the BBMap v39.01 bioinformatic pipeline (75). Briefly, common sequencing adaptors, artifacts, phix, and human sequences were removed followed by error correction. The trout genome GCA_013265735.3 obtained from the NCBI database (Downloaded 5th December 2023) contained both trout chromosomes and other assembled contigs. Only trout chromosomes were retained, as other contigs were identified as bacterial contaminates in origin through CheckM v1.2.2 (76). BBMap was used to map reads against trout chromosomes to identify the proportion of reads identified as trout. Reads that were not mapped to the trout chromosomes were assembled into contigs using metaSPAdes v3.15.5 (77) without error correction, as this had already been performed. MetaBAT2 v2.12.1 (78) and jgi_summarize_bam_contig_depths function were used to bin metagenomes into MAGs, where required mapping files of assembled metagenomes were mapped back to non-trout reads from other samples within the same treatment group using BBMap. CheckM was used to taxonomically identify MAGs obtained from MetaBAT2 with associated completeness and contamination scores.

## RESULTS

### Summary of host depletion methods DNA concentration and quality

A total of 87 individual fish (across the three sampling dates) were swabbed and processed for application to the five different eukaryotic DNA depletion methods tested, together with a standard DNA extraction as a benchmark to assess DNA depletion efficacy. The concentration and purity of the extracted DNA, applying the different protocols, are shown in Table 1.

### All host depletion methods increase the proportion of 16S rRNA to 18S rRNA

To assess the efficacy of host (fish) DNA depletion for various host depletion methods, we employed qPCR methods to calculate the proportion of fish to bacterial DNA concentration (Table 2; Fig. 1). All host depletion methods had significantly higher log-fold ratios of 16S to 18S rRNA gene compared to the control. This indicated a reduction in host (18S) DNA and corresponded to a greater proportion of prokaryotic material. In order of 16S enrichment, the most effective method was the Zymo method, followed by the Molysis kit method, CpG-methylated method, Hypotonic Lysis method, and lastly, the Hypotonic Lysis method using PBS (Table 2; Fig. 1).

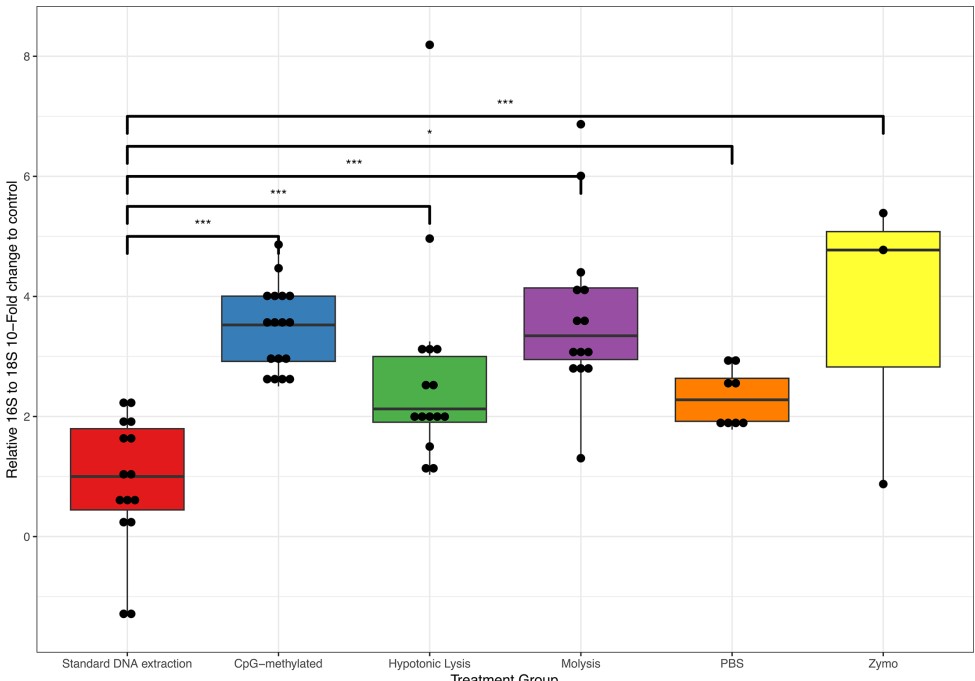

**FIG 1** Relative amount of 16S to 18S rRNA gene retained after each DNA depletion method expressed as 10-fold change. All host depletion methods are compared against the standard DNA extraction control. *$P \leq 0.05$ and ***$P \leq 0.001$.

## Host depletion methods do not reduce the abundance of mitochondrial ASVs

16S amplicon sequencing was performed to assess whether different DNA depletion techniques depleted microorganisms differently from those present in the untreated sample, and thus resulted in a biased view of the microbiome. In the 16S amplification (after quality control and contaminant removal), there was a median of 82,030 (26,762; 178,032, 5th and 95th percentiles, respectively) ASVs obtained per sample comprising 5,001 taxa. As host cells contain mitochondria, and due to their close homology to the 16S bacterial rRNA gene, are PCR amplified, we assessed the effectiveness of host depletion techniques to remove host mitochondrial ASVs before sequencing. A lower abundance of mitochondria sequences may indicate effective host cell DNA removal before sequencing. The relative abundance of ASVs identified as mitochondrial and grouped by treatment group was not significantly different from control samples, suggesting no difference in the proportion of mitochondrial ASVs in each treatment group (Fig. 2; Table 3).

## All host depletion methods have similar alpha diversity metrics

Alpha diversity metrics for the extractions from different host depletion-treated groups were not significantly different compared with control DNA extraction groups (Shannon, Chao1, or Faith's phylogenetic diversity [PD]), indicating no change in richness, evenness, or distribution of taxa between them (Fig. 3; Table 4).

## The Molysis host depletion method results in an unrepresentative beta diversity

When calculating beta diversity, samples were partitioned by sampling date (given the dynamic nature of skin microbiome), and beta diversity metrics were calculated separately for each sampling date, as random effects could not be included in associated PERMANOVA statistical tests (Fig. S2). Only the Molysis treatment group showed a statistically significant different beta diversity compared to the control group (01 Dec 2022, $R^2$ = 23%, adjusted $P$-value = 0.035), indicating retention of different taxa

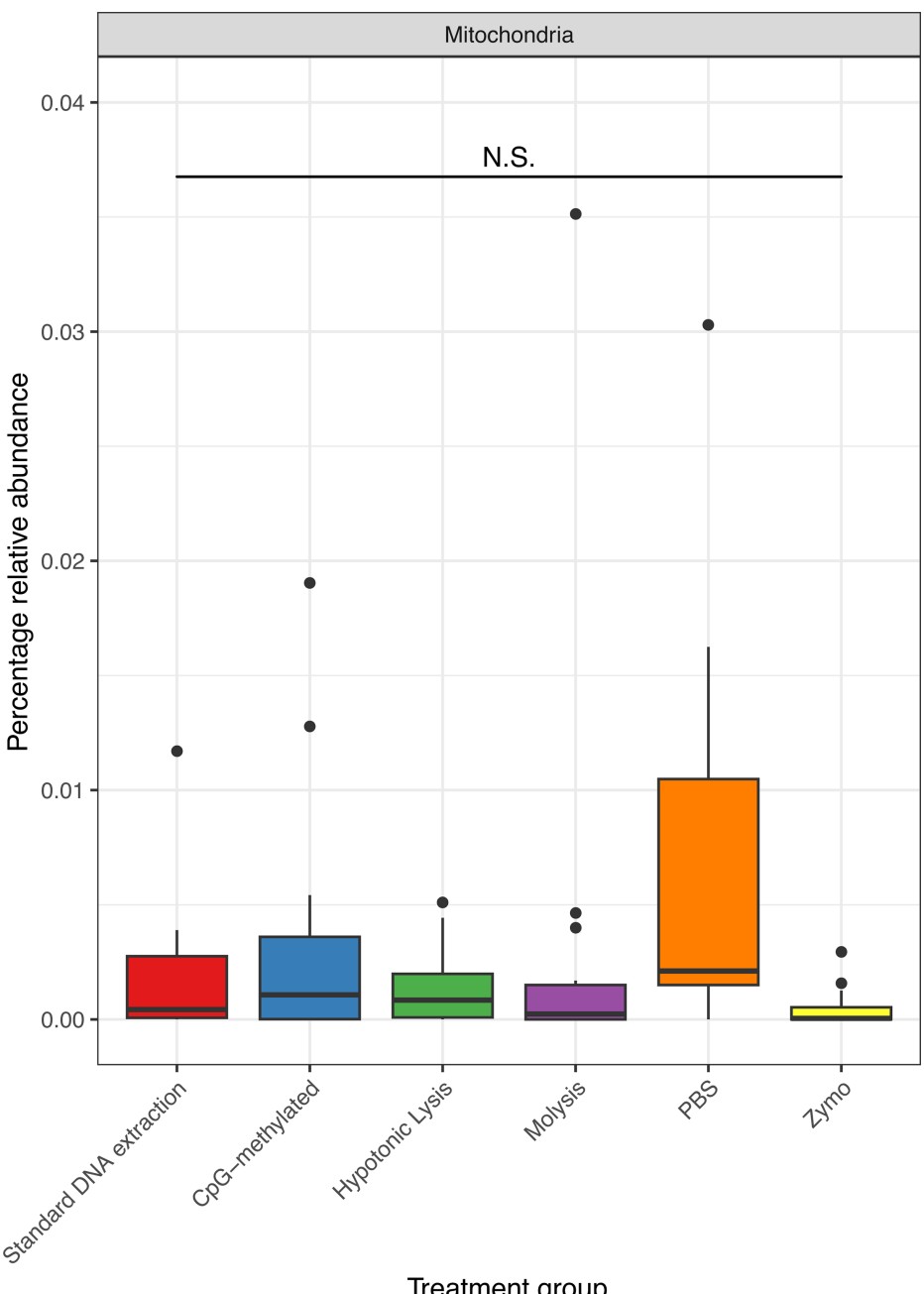

**FIG 2** Relative abundance (as a percentage) of 16S ASVs identified as mitochondrial grouped by different host depletion methods and a standard (control) DNA extraction method. N.S. = not significant.

compared to the control (Fig. S3 and S4; Table S2). The Molysis treatment group was also the only treatment group that significantly differed from all other treatment groups for the December sampling, and also from the Hypotonic Lysis and Zymo treatment groups for the November sampling (Fig. S3).

When all sampling dates within a treatment group were pooled, the Molysis group was also significantly different from the control extraction when assessed using a Binary Jaccard (similar to presence-absence), Weighted-UniFrac (inclusion of phylogenetic distance in beta-diversity calculations), Aitchison's distance (normalizing and scaling beta-diversity by ASV), and a Bray-Curtis dissimilarity matrix. This approach for pooled samples across the collection periods also revealed differences in the Zymo method

**TABLE 3**  Application of a linear model (adjusted for multiple hypothesis testing) to assess different host depletion methods' based on the relative abundance of mitochondrial ASVs compared to a standard (control) DNA extraction method[a]

| | Comparison | Estimate (difference from control) | Std. error | t-value | P-adjusted value | Significance |
|---|---|---|---|---|---|---|
| | CpG-methylated | 0.002 | 0.002 | 0.904 | 0.615 | N.S. |
| | Hypotonic lysis | 0 | 0.002 | −0.143 | 0.887 | N.S. |
| Standard DNA extraction | Molysis | 0.002 | 0.002 | 0.992 | 0.615 | N.S. |
| | PBS | 0.006 | 0.002 | 2.36 | 0.104 | N.S. |
| | Zymo | −0.001 | 0.002 | −0.6 | 0.688 | N.S. |

[a]N.S. = not significant; *P* > 0.05. Formula "mitochondria.abundance − Treatment.group + (1 | Sampling.date)".

when analyzed using a Bray-Curtis or Binary Jaccard dissimilarity matrix, and in the Hypotonic Lysis method when distances are calculated using the Bray-Curtis dissimilarity matrix. Overall, this supports an unbalanced microbial community recovery from fish skin in all tested beta diversity metrics for the Molysis method, but is also a feature of depletion using Zymo and the Hypotonic Lysis methods (Fig. 4).

## Over-representation of Proteobacteria and/or under-representation of Bacteroidota and Bacilli taxa

Using differential abundance, we determined which taxa across the different phyla and classes accounted for changes in microbial communities extracted using the different host depletion treatments compared with the control. The bacterial class Bacilli and order Mycoplasmatales were significantly depleted in the CpG-methylated, Molysis, and Zymo extractions. The Firmicutes phyla were also depleted through the use of the Zymo kit extraction. The Bacteroidota phylum, Bacteroidia class, and Flavobacteriales order were also depleted in the Molysis treatment group. In contrast, the Proteobacteria phylum, Alphaproteobacteria class, and Rhizobiales order were significantly enriched in the Molysis treatment group, whereas only the Proteobacteria phylum was significantly enriched in the Zymo group (Fig. 5; Table S3). The hypotonic lysis and PBS methods did not result in any significantly differentially abundant taxa compared with the standard DNA extraction method.

## Host depletion techniques only result in small increases in microbial DNA

Applying metagenomics to the standard DNA extraction, CpG-methylated, and Zymo methods, we determined the ratio of nucleotides identified as rainbow trout from a reference rainbow trout genome (Fig. 6). Raw metagenomes provided an average of

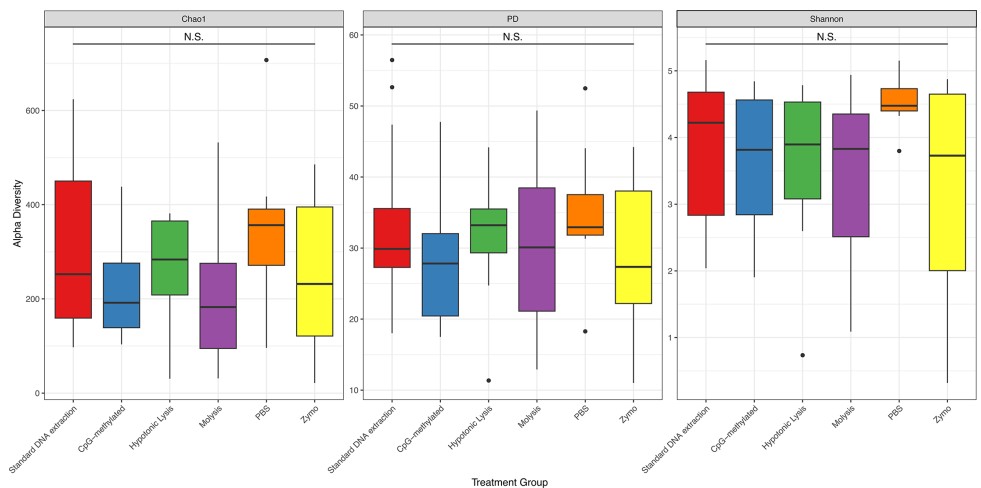

**FIG 3**  Alpha diversity (Chao1, Fath's phylogenetic diversity [PD], and Shannon) of different host depletion treatments on rainbow trout skin swabs compared to a standard (control) DNA extraction. N.S. = not significant.

**TABLE 4** Phylogenetic diversity alpha diversity (Shannon, Chao1, and Fath's) after the application of different host depletion treatment groups compared with a control DNA extraction group, tested using a linear model adjusted for multiple hypothesis testing[a]

| Alpha diversity metric | Comparison | | Estimate (difference from control) | Std. error | t-Value | P-adjusted value | Significance |
|---|---|---|---|---|---|---|---|
| Shannon | | CpG-methylated | −0.10 | 0.29 | −0.33 | 0.84 | N.S. |
| | | Hypotonic lysis | −0.15 | 0.30 | −0.49 | 0.80 | N.S. |
| | | Molysis | −0.23 | 0.31 | −0.73 | 0.71 | N.S. |
| | | PBS | −0.02 | 0.39 | −0.06 | 0.95 | N.S. |
| | | Zymo | −0.56 | 0.30 | −1.87 | 0.20 | N.S. |
| Chao1 | | CpG-methylated | −87.03 | 40.47 | −2.15 | 0.16 | N.S. |
| | Standard DNA extraction | Hypotonic lysis | −30.42 | 41.91 | −0.73 | 0.71 | N.S. |
| | | Molysis | −80.41 | 42.60 | −1.89 | 0.20 | N.S. |
| | | PBS | −12.80 | 53.46 | −0.24 | 0.86 | N.S. |
| | | Zymo | −52.91 | 41.18 | −1.28 | 0.41 | N.S. |
| Faith's phylogenetic diversity | | CpG-methylated | −4.89 | 2.83 | −1.73 | 0.20 | N.S. |
| | | Hypotonic lysis | −1.78 | 2.93 | −0.61 | 0.76 | N.S. |
| | | Molysis | −3.54 | 2.98 | −1.19 | 0.43 | N.S. |
| | | PBS | −1.47 | 3.74 | −0.39 | 0.84 | N.S. |
| | | Zymo | −5.11 | 2.88 | −1.78 | 0.20 | N.S. |

[a]N.S. = not significant. Formula "alpha.div − Treatment.group + (1|Sampling.date)".

22,581,152 paired-end reads per sample. Use of both the CpG-methylated and Zymo treatment groups resulted in a lower number of trout nucleotides by 3.66%; $P = 0.015$ and 2.31%; $P = 0.059$ (N.S.), respectively, for which only the CpG-methylated group was statistically significant (Table 5). We attempted to construct MAGs from each of the treatment groups using both an individual sample and a within-treatment co-assembly method; however, both were unsuccessful in obtaining any MAGs that were not taxonomically identified as trout, possibly due to the high numbers of trout reads still present even after depletion.

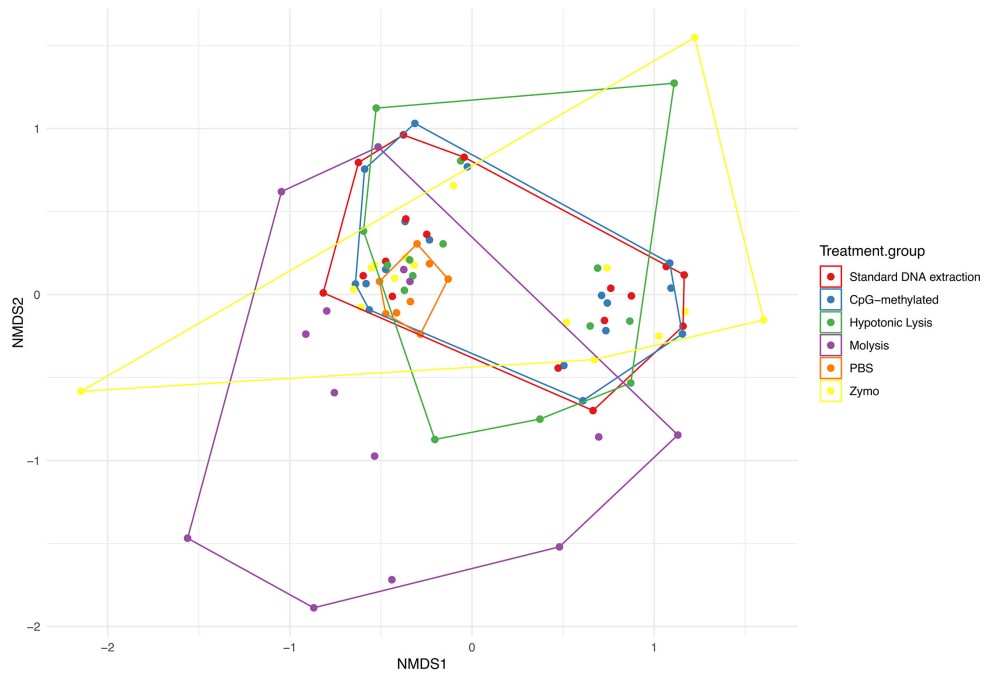

**FIG 4** Beta-diversity calculated from a Bray-Curtis dissimilarity matrix aggregated at a genus level on compositional data (relative abundance) grouped by host depletion treatment group, visualized using an NMDS ordination.

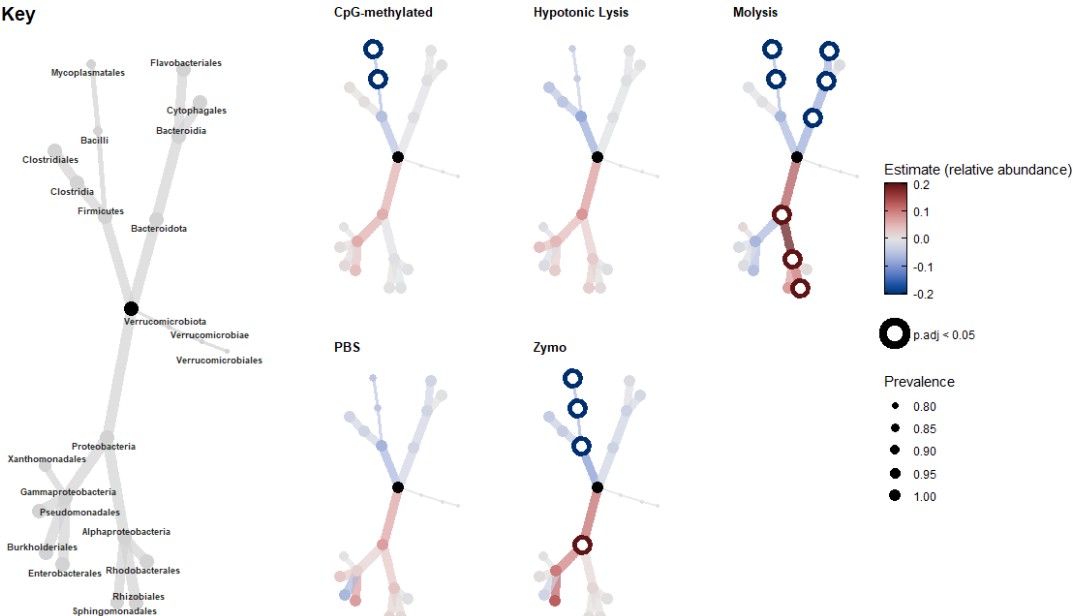

**FIG 5** Differential abundance of bacteria at a phylum, order, and class level compared to control DNA extractions as determined by a linear model corrected for multiple hypothesis testing. Taxa were considered significantly differentially abundant at a *P*-adjusted < 0.05 threshold and highlighted in bold. The degree of differential abundance is shown in color, with red indicating enrichment and blue depletion compared to controls. The size of tree nodes indicates bacterial taxa prevalence in all samples from each host depletion treatment group. A taxa key provides taxonomic identification of each tree node.

## DISCUSSION

Here, we assess the effects of five different host depletion methods for their ability to reduce the proportional amount of host DNA without biasing the recovery of microbial DNA for trout skin mucosal microbiomes. 16S rRNA to 18S rRNA gene ratios assessed using qPCR suggested all methods were effective at reducing the proportion of host DNA, but the relatively small increases in the percentage non-host reads (e.g., 3.7% and 2.3% for the CpG-methylated and Zymo methods, respectively) as determined via metagenomics are unlikely to be practically beneficial. Notably, the Molysis method resulted in a significant difference in bacterial community composition compared to untreated controls. We did not find that any of the extraction methods affected the alpha diversity metrics. This suggests that alpha diversity may not be a sufficiently sensitive measure for detecting taxonomic differences between host depletion techniques, particularly as significant differences in taxonomic differential abundance were observed for the Zymo, Molysis, and CpG-methylated-based treatments. Considering the limited enrichment and potential for bias, employing extra sequencing depth for untreated skin microbiome extracts may currently represent a more cost-effective approach than the application of host depletion kits.

### Effective host DNA depletion methods are limited to certain biological sample types

Previously reported host DNA depletion methods have achieved reductions of up to 80% of host DNA, but this differs considerably, in part depending on the sample type. For example, application of the lyPMA method (hypotonic lysis followed by propidium monoazide treatment) to milk samples was similarly observed to provide minimal increases in bacterial or unclassified reads (below 1% and 3%, respectively; Ganda et al., (14), as we found here for our fish skin work. This limited depletion contrasts with the effectiveness of the lyPMA method applied to the removal of host DNA from human saliva (24) (from 89% to ~8.5%) and the Benzonase 1 and 2 methods, analogous to our Hypotonic Lysis method, that has been shown to lead to the depletion of human reads

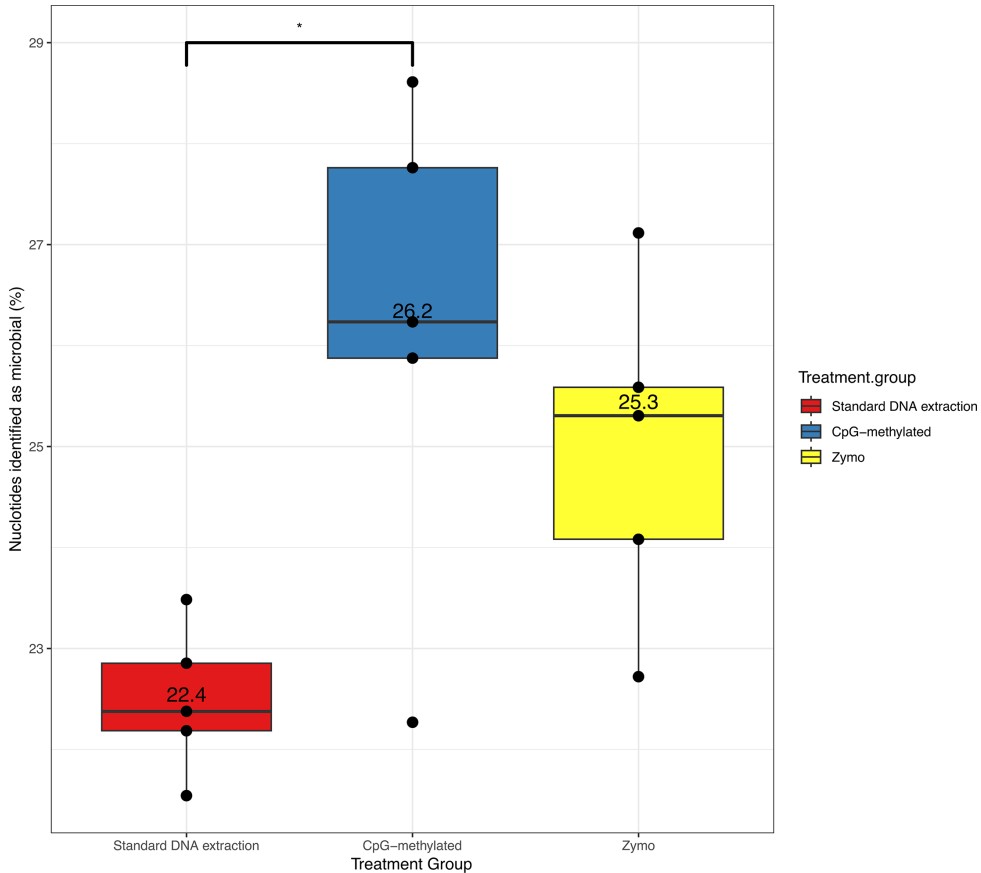

**FIG 6** Relative abundance of nucleotides from trout skin swabs not identified as rainbow trout (and therefore likely of microbial origin) from five randomly selected samples in treatment and control groups.

in human sputum samples by 30% to 37% (25). These comparisons suggest that the hypotonic lysis method, suitable for removing human cells from saliva and sputum, is less effective for samples with buffering capacity against hypotonic changes, such as milk or fish skin epithelial cells (especially from anadromous species like rainbow trout), which are likely more resistant to osmotic stress and chemical lysis (14, 79).

Fish skin mucus has a complex biochemical composition rich in glycosylated proteins (mucins), lipids, and other antimicrobial proteins, which may act as a barrier protecting host cells from lytic reagents (80). This, in turn, may impede the diffusion of the lysis agents into host cells, hinder the access of nucleases employed to successfully degrade lysed host cells, and/or quench the reaction of these agents before they can exert their effects on host DNA. These barrier effects are likely to be more pronounced in fish mucosa compared to the less viscous human saliva fluids, perhaps helping to explain, at least in part, their success when applied in human samples, but not in fish samples (80).

Fish cells also likely exhibit greater resistance to osmotic stress compared to mammalian cells, particularly for species such as rainbow trout that are anadromous, moving between fresh and salt waters as part of their natural life cycle, and where their skin is in constant interface with aquatic environments with their varying levels of salinity

**TABLE 5** Linear model testing (corrected for multiple hypotheses) on the relative abundance of nucleotides from trout skin swabs not identified as trout for each treatment group versus an untreated control group[a]

| Comparison | | Estimate | Std. error | t-value | P-adjusted value | Significance |
|---|---|---|---|---|---|---|
| Standard DNA extraction | CpG-methylated | 3.66 | 1.05 | 3.48 | 0.01 | * |
| | Zymo | 2.32 | 1.06 | 2.18 | 0.06 | N.S. |

[a]Formula "microbial.perc − Treatment.group + (1|Sampling.date)" *$P < 0.05$, N.S. = not significant.

(79). Indeed, fish skin epidermis cells possess specialized osmoregulatory mechanisms and cell membrane compositions (lipid profiles, aquaporin expression) that likely provide robust protection against the hypotonic cell lysis that is effective for human cells (81, 82).

In the case of the NEBNext Microbiome DNA Enrichment kit, this relies on capturing CpG-methylated DNA, more common in eukaryotic genomes than in bacterial genomes. However, CpG-methylated island density and distribution vary across taxa (83, 84). Rainbow trout and other fish species have relatively poor gene promoters in CpG islands (<1%) compared to mammalian promoters (~60%), suggesting that these CpG methylated sites in trout genomes are patterned differently than in mammals for which this kit was primarily developed. This potentially limits the methyl-binding protein capture system's effectiveness (85). This likely contributed to the modest performance in fish skin cells of only a 3.66% increase in non-host reads. Other methods could include a CRISPR (Jumpcode) method or a CHIP pulldown method approach (86, 87).

## Introduction of bias in the bacterial community composition using the Molysis method

We observed a distinctly different beta diversity profile in samples treated with the Molysis method compared with other treatments, which has also been shown for human saliva samples (24). Contrasting with these analyses, various other authors working on human sputum, milk, and tissue samples have reported similar abundances of taxa when applying the Molysis treatment method (25, 29, 39); however, these studies were not analyzed statistically and relied on visual differences. A bias against gram-negative bacteria in milk samples was also observed when using the ly-PMA approach by Ganda et al. (14), showing that different depletion chemistries can selectively affect different microbial groups. The Molysis and benzonase methods also resulted in altered community structure in human sputum samples, which was attributed to the partial degradation of environmental DNA (eDNA), particularly from high-eDNA producers such as *Pseudomonas aeruginosa* (25). We speculate that in our study, taxa within the fish skin microbiome may produce eDNA that contributes to the apparent depletion of these taxa. DNA from the eDNA-producing taxa is degraded by nuclease-based methods not used in our controls before sequencing, leading to a false positive in apparent depletion levels (25, 88). Collectively, these studies suggest that the Molysis/Molzym method introduces a bias when assessing the bacterial community composition, resulting in significantly different profiles compared to controls.

## Differential abundance analyses reveal biases in the bacterial compositions in samples for host-depleted treatments

Using differential abundance analysis, we found a bias in bacterial taxa following host depletion. In particular, the Firmicutes phyla (synonym Bacillota) were significantly depleted in the CpG-methylated, Molysis, and Zymo treatment groups. Both the Zymo and Molysis methods rely on the lysis of host cells before DNA extraction, and it is possible that the chaotropic solutions, designed to lyse host cells, may also degrade bacteria from the Firmicutes phylum. This is unexpected as Firmicutes are generally gram-positive with a thick peptidoglycan layer (89), conferring resistance to harsh conditions, and thus would likely be more difficult to degrade (90).

The enrichment of Proteobacteria (predominantly gram-negative [91]) when using the Molysis and Zymo methods may possibly be related to the improved cell lysis or DNA extraction of these taxa. In turn, this may suggest these methods are better at or assist DNA extraction methods in lysing these taxa, possibly due to the weakening of the cell wall via its chaotropic buffer. We found no evidence for a trend suggesting that hard-to-lyse bacteria, such as gram-positive bacteria, were overall more prevalent in host depletion-treated microbiomes, suggesting these proprietary methods are not affected by an additional bacterial peptidoglycan protective layer. The enrichment of Proteobacteria taxa in the Zymo and Molysis methods, however, does suggest an improved DNA extraction ability, indicated by their higher recovery of bacterial DNA from these taxa.

However, these observed biases may not solely stem from differential efficacies in cell lysis. Post-lysis steps within each method, such as DNA degradation by nucleases, may have off-target effects on certain bacterial DNA structures, differential bindings, and be washed or eluted more efficiently, which could contribute to observed differences (25, 92). For instance, if DNA from lysed Firmicutes is more susceptible to nucleases present or is less efficiently recovered during purification steps due to residual reagents or additional washing steps in each method, this would appear as a depletion bias, irrespective of cell wall robustness. The ecological implications of these biases could misrepresent fish skin microbiomes' compositions, affecting interpretations of microbiome health, dysbiosis, and functional potential, such as the underestimation of Firmicutes and the overestimation of Proteobacteria.

## Interpreting metagenomic results and MAG limitations

The modest metagenomic improvements of CpG-methylated (3.66%) and Zymo (2.31%) of non-host DNA highlight the limited practical utility of these methods for microbial enrichment. These findings are supported by Pereira-Marques et al. (26), who demonstrated that high proportions of host DNA compromised the sensitivity of detecting rare microbial species and reduced the accuracy of relative abundance estimates, particularly when sequencing depth was insufficient. The failure to construct any MAGs even in host depletion samples is likely a direct consequence of the persistently high percentage of host DNA. For example, a 3.66% reduction from the estimated 99% host DNA still leaves over 95% host DNA in a sample, levels that Pereira-Marques et al. (26) show are problematic for microbiome characterization and MAG recovery. If depletion methods are largely ineffective, introduce bias, and result in the loss of total DNA (this study showed an average of a 10-fold decrease), then the deeper sequencing of untreated samples despite the higher proportion of host reads might be more reliable, albeit a more expensive strategy for the accurate microbial community profiling and MAG recovery from fish skin mucus.

We note, however, that the failure to recover MAGs is not solely the function of the percentage of remaining host DNA but is also linked to the absolute amount and quality of DNA sequenced (93). If host depletion methods result in a loss of fragmentation of microbial DNA (as suggested in reference [14]) for some kits in milk and the lower total DNA yields within this study, the remaining microbial DNA may be insufficient in terms of coverage depth across individual genomes or too fragmented for successful assembly and binning into MAGs, even if the relative proportion of host DNA has slightly decreased.

## Research needs for enriching prokaryotic DNA for studies on fish skin microbiomes

The findings from this study clearly show the need for new and/or improved methods optimized for host depletion strategies, specifically tailored for fish skin mucus and potentially other challenging mucosal samples. This study focused on juvenile rainbow trout (*Oncorhynchus mykiss*) as a representative fish model; however, the host depletion methods evaluated target conserved host-cell features (e.g., CpG islands, mucosal surfaces) that are broadly shared across teleost fish, suggesting the given findings are likely applicable to wider fish species. Nevertheless, further validation in species with different ecological traits (habitat, diet, life stage) will be important, as there will inevitably be some differences in their skin (and other) microbiomes as they have adapted to inhabit diverse aquatic environments.

Future research might usefully explore mucus disruption pre-treatments, where the enzymatic or chemical breakdown of the mucus matrix would improve accessibility of host cells to DNA-depleting reagents (94). Given the potential resilience of fish skin cells to lysis protocols (exemplified here for rainbow trout as an anadromous fish species), different lysis conditions and reagents should be explored. These might include the efficacy of detergents, enzymes, or other physical methods optimized for fish epidermal

cells. Additionally, strategies that do not rely on CpG-methylated patterns, which can differ drastically based on host species (i.e., between fish and mammals), should be investigated. This may include strategies to target specific fish epidermal properties or genomic sequences to aid host cell depletion (95).

We also highlight the efficacy of validation metrics on host depletion, such as qPCR and alpha diversity, as ineffective at determining the efficacy of host depletion methods. While qPCR can indicate the relative reduction in host DNA, it may not reflect biases introduced or the suitability of the DNA for downstream applications. As shown here, alpha diversity remained unchanged despite significant shifts in community composition detected by differential abundance analysis. Instead, efforts toward shotgun metagenomic sequencing outputs, including statistically supported changes in relative abundance of taxa and the potential for MAG recovery, should be prioritized as these have been shown to be more effective indicators of host depletion method.

Other important areas to consider are the absolute microbial load of samples after going through the host depletion method(s). Quantitative microbial load assessments, such as total 16S copy numbers, are required as depletion techniques may result in the loss of microbial cells or DNA, which cannot be captured by relative abundance data alone. The total microbial DNA in a fish skin swab is low (less than 1%), and further reduction may reduce microbial DNA concentrations below levels effective for shotgun metagenomic and amplicon methods. The consequences of this may include off-target amplification in 16S studies (such as an increased ratio of mitochondria or chloroplasts from host cells) and/or require further enrichment, such as amplification before shotgun metagenomic sequencing, introducing bias. Lastly, host depletion methods should consider the contributions of both host and microbial eDNA in assessing the efficacy of host depletion techniques. eDNA may be easily removed during host depletion steps, such as washing and endonuclease degradation, skewing the metrics assessing the effectiveness of host depletion.

To conclude, a successful host depletion approach for fish generally will likely require a multi-faceted approach that addresses: (i) the efficient disruption of the mucosal barrier; (ii) effective and robust fish-specific cell lysis; (iii) comprehensive degradation or removal of both fish nuclear and mitochondrial DNA; and (iv) robust preservation of the diversity and integrity of microbial cells and their DNA. Developing such methods would help address our understanding of the functional capacity of these microbiomes crucial for research in aquaculture and environmental health. Effective host depletion methods would enable researchers to move beyond taxonomic cataloging and explore changes in gene composition, providing a more direct insight into the functional responses of the microbiome to perturbations. Improved host depletion techniques would benefit not only metagenomic studies but also transcriptomic, metabolomic, and proteomic analyses by enriching for taxa of interest and reducing host DNA.

## ACKNOWLEDGMENTS

The authors would like to thank Dr. Jamie McMurtrie and Dr. Sanjit Debnath for their help and advice throughout this experiment.

A.G.B. was supported by the FRESH—NERC Center for Doctoral Training in Freshwater Biosciences and Sustainability (GW4 FRESH CDT) [NE/R011524/1] (2401467)

Conceptualization was conducted by A.G.B. and C.R.T. Methodology was developed by A.G.B., J.C., B.T., and C.R.T. Software analysis was provided by A.G.B. Formal analysis was carried out by A.G.B. The investigation was conducted by A.G.B. Resources were provided by B.T. and C.R.T. Data curation was performed by A.G.B. The original draft was written by A.G.B., and all authors (A.G.B., J.C., B.T., and C.R.T.) contributed to the review and editing of the manuscript. Visualization was undertaken by A.G.B. Supervision was provided by J.C., B.T., and C.R.T. Project administration was managed by A.G.B., B.T., and C.R.T., while C.R.T. led funding acquisition.

## AUTHOR AFFILIATIONS

[1]Biosciences, Faculty of Health and Life Sciences, University of Exeter, Exeter, Devon, United Kingdom
[2]Sustainable Aquaculture Futures, University of Exeter, Exeter, Devon, United Kingdom
[3]School of Biosciences, Cardiff University, Cardiff, United Kingdom

## AUTHOR ORCIDs

Ashley G. Bell  http://orcid.org/0000-0003-0198-2205
Ben Temperton  http://orcid.org/0000-0002-3667-8302

## FUNDING

| Funder | Grant(s) | Author(s) |
| --- | --- | --- |
| Natural Environment Research Council | [NE/R011524/1] (2401467) | Ashley G. Bell |

## AUTHOR CONTRIBUTIONS

Ashley G. Bell, Conceptualization, Data curation, Formal analysis, Investigation, Methodology, Project administration, Software, Visualization, Writing – original draft, Writing – review and editing | Jo Cable, Methodology, Supervision, Writing – review and editing | Ben Temperton, Methodology, Project administration, Resources, Supervision, Writing – review and editing | Charles R. Tyler, Conceptualization, Funding acquisition, Project administration, Resources, Supervision, Writing – review and editing

## DATA AVAILABILITY

The data for this study have been deposited in the European Nucleotide Archive (ENA) at EMBL-EBI under accession number PRJEB82663. All bioinformatic processes are available at: https://github.com/ash-bell/host_depletion.

## ETHICS APPROVAL

All animal work was carried out in accordance with the EU Directive for the protection of animals used for scientific purposes (2010/63/EU) and the UK Animals Scientific Procedures Act (ASPA) 1986.

## ADDITIONAL FILES

The following material is available online.

### Supplemental Material

**Supplemental material (Spectrum01838-25-s0001.docx).** Figures S1 to S4; Tables S1 to S3.

### Open Peer Review

**PEER REVIEW HISTORY (review-history.pdf).** An accounting of the reviewer comments and feedback.

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
