## [Reviewer comments · Microbiology Spectrum]

Microbiology Spectrum

Assessment of the Effectiveness of Host Depletion Techniques for Profiling Fish Skin Microbiomes and Metagenomic Analysis

Ashley Bell, Jo Cable, Ben Temperton, and Charles Tyler

Corresponding Author(s): Ashley Bell, University of Exeter

Review Timeline:

Submission Date:	June 16, 2025
Editorial Decision:	September 4, 2025
Revision Received:	October 17, 2025
Accepted:	November 24, 2025

Editor: Justine Debelius

Reviewer(s): The reviewers have opted to remain anonymous.

Transaction Report:

DOI: <https://doi.org/10.1128/spectrum.01838-25>

Re: Spectrum01838-25 (Assessment of the Effectiveness of Host Depletion Techniques for Profiling Fish Skin Microbiomes and Metagenomic Analysis)

Dear Dr. Ashley G Bell:

Thank you for the privilege of reviewing your work. Below you will find my comments, instructions from the Spectrum editorial office, and the reviewer comments.

Both reviewers thought the manuscript had merit and represented a contribution to the field, however, they identified several issues. The reviewers suggested a better description of sample replication and how that replication potentially affected the analysis. Please pay careful attention to similarity within replicates and consider how this might influence analytical results. Additionally, they raised concerns about how the ratio of bacteria call counts to fish counts were compared.

Revision Guidelines

Sincerely,
Justine Debelius
Editor
Microbiology Spectrum

Reviewer #1 (Public repository details (Required)):

All sequencing data needs to be public

Reviewer #1 (Comments for the Author):

• Title: Assessment of the Effectiveness of Host Depletion Techniques for Profiling Fish Skin Microbiomes and 4 Metagenomic Analysis

The authors address a very important problem in fish (and all vertebrate) microbiome research: the presence of host DNA in host-associated samples. The majority of methods developed have been demonstrated for humans but have not been tested in fish. While the overall results of this paper are 'negative', the results are still very important to communicate since it can save other's time and money on these methods while focusing on development of new methods. Overall the analyses and design are robust. I offer a few suggestions to improve the ms.

The authors specifically attempt 2 methods: the CpG approach which relies on the fact that host DNA (vertebrates) is methylated more so than bacteria. They also attempt a selective lysis method. My overall feedback is that there are several other approaches out there which the authors did not consider including using a CRISPR approach (Jumpcode), [<https://www.jumpcodegenomics.com/applications/microbiome>], PMA [PMID: 29482639] and CHIP pull down PMID: 39982577. Authors need to include these approaches in the discussion as alternative approaches to be tested in the future.

Regarding the negative result of selective host depletion, can the authors comment or speculate on the idea that perhaps all of the bacteria in the skin mucus is already lysed (just extracellular DNA)?

Line 171: How were samples stored overnight if extractions were the next day? How did you ensure that DNases were not activated and degrading microbial DNA?

Line 251:
"All qPCR reactions"

RESULTS

Line 434: Regarding the qPCR assay, I am concerned here about off target amplification and therefore a distorted signal: How do the authors know that the 18S target which was targeted and amplified from the fish was in fact from the fish? How do they know if the 16S target was actually 16S and not also amplifying 12S (mitochondria) which is very common? Their bacteria ratio might be inflated. I would strongly suggest running some of these qPCR reactions through shotgun metagenomics to verify the ratios of what is actually amplifying. This could have serious implications for what's happening. Alternatively, I would have considered using a more specific marker for the fish. This could be a marker which is very specific to the fish genome include a repeat region (microsat).

Figure 1: Could the authors display this another way: maybe a simple stacked bar of (% 16S, % 18S)? As of now it's a bit confusing as to what I'm looking at. You can keep the existing figure but maybe have a separate panel so its more intuitive.

Line 458: I am concerned about this conclusion of using mito ASVs as an endpoint. While 16S EMP primers do amplify 12S, its not the intended target. I think this is a reasonable thing to report but I don't believe you can make the argument that there are less mito overall and an inefficiency in host depletion. If authors really want to address this, they should design a specific mito DNA assay to amplify (e.g. qPCR). They could also do a rainbow trout specific marker. Check the RTqPCR literature and I'm sure there are primers out there for house keeping genes.

Line 488

Figure 4:

These are some pretty highly variable numbers for alpha diversity. Were the statistical analyses used considering pairwise comparisons to control for the variation in alpha across biological samples? If not, that may be why there aren't statistical differences (biological variation is greater than kit variation). Please update

Line 496: Again, does this mean the same samples were processed in the control group along with all of the experimental groups? If not, the authors need to thoroughly describe how their samples used from the various kits may have differed and thus influenced the diversity metrics. For instance, were the fish used in the Molysis simply collected a different day or from a different tank? All of these minute factors can influence the fish microbiome.

Line 523: Mycoplasma are likely intracellular here.

• ,

Reviewer #2 (Public repository details (Required)):

Authors mentioned that data have been deposited in the European Nucleotide Archive (ENA)767 at EMBL-EBI under accession

Reviewer #2 (Comments for the Author):

Bell et al. investigated different host DNA depletion techniques for profiling the skin microbiota of fish, using a freshwater species (rainbow trout) as a model organism. Several methods were applied to reduce host DNA contamination. This work is particularly interesting from a metagenomics perspective, as this approach stands to benefit significantly from effective host depletion. I found the study interesting and potentially useful, but it lacks critical information in several sections. My main concerns are 1) overgeneralization: The authors make general statements about applicability to "fish" based on results from a single species (juvenile rainbow trout). However, fish species may vary significantly in skin mucosa characteristics depending on habitat (freshwater vs saltwater), feeding strategies (herbivory vs carnivory), life stage (juveniles vs adults), and other ecological traits. These factors could influence how well host depletion methods work. The authors should clearly state that the study used a farmed rainbow trout and ensure the species name is included in the title, abstract, and throughout the manuscript. Although the primary focus is on methodology, the biological context is highly relevant and should not be overlooked. Some ecological background on the species used would be beneficial. 2) Experimental design details: The manuscript lacks clarity on key aspects of the experimental setup: How many samples and replicates were used per fish? Were individual fish sampled multiple times (e.g., multiple swabs per fish), or were different fish used for each method? Were the swabs tested using multiple methods, or were separate swabs collected for each method? Individual variability between fish could influence the results, so it's important to report and acknowledge these details. 3) method selection rationale: were the five depletion methods selected based on prior success in other organisms? If so, which ones e.g. humans, marine animals, or other fish? Have these methods been tested before on mucosal layers from any aquatic animals?

The section introducing the species and methodologies needs to be more clearly written. Consider revising the start of the methods section to clarify the sampling strategy and add a paragraph in the introduction with background on the fish species.

Specific comments:

Lines 76-78 and 96-100 are missing appropriate references.

Lines 147-168: It's unclear how many fish were sampled and why euthanasia was necessary if only skin mucosa was collected. Please clarify. Also, what are the sampling time points referenced, and why are they relevant?

Line 177 suggests that all samples were extracted using the same CTAB/EDTA/chloroform method, but line 251 shows that different DNA extraction methods were used for each host depletion strategy. Please clarify which extraction protocol was used for each method and ensure consistency/clarity throughout the manuscript.

Line 427: Why do the sample numbers ("n=") in Table 1 differ between methods? Were these from the same fish sampled multiple times or from different fish? Clarify the sampling design.

Line 438: Authors mentioned Figure 1, then Table 2. Figure 1 should be introduced before Table 2 if referenced earlier.

Line 488: Differences shown in Figure 3 could be due to multiple factors. This goes back to my comment regarding the methods. It is essential to clarify the number of samples used and how they were collected to support the findings.

Line 637: Using "eDNA" may be confusing, as this typically refers to environmental DNA in the broader literature.

Lines 640-645: Is this speculation? References or clarification to support this would help.

Lines 650-657: This section is difficult to follow. Please rephrase for clarity.

Lines 687-692: DNA from mucosa can be challenging to extract. Was the DNA integrity assessed (e.g., using a 0.8-1% agarose gel to check for high molecular weight DNA)? DNA fragmentation could explain low bacterial DNA recovery in MAGs.

Please refer to the track changes document for accurate line numbers, thank you

Reviewer #1

S/N	Comment	Response
1	All sequencing data needs to be public	Thank you for your comment. All sequencing data have been made publicly available in the European Nucleotide Archive (ENA) under accession number PRJEB82663.
2	The authors specifically attempt 2 methods: the CpG approach which relies on the fact that host DNA (vertebrates) is methylated more so than bacteria. They also attempt a selective lysis method. My overall feedback is that there are several other approaches out there which the authors did not consider including using a CRISPR approach (Jumpcode), [https://www.jumpcodegenomics.com/applications/microbiome], PMA [PMID: 29482639] and CHIP pull down PMID: 39982577. Authors need to include these approaches in the discussion as alternative approaches to be tested in the future.	Thank you for this suggestion. We have now included the CRISPR-based approach and the CHIP pulldown method in the discussion (lines 606–607). The lyPMA approach was already cited (line 568).
3	Regarding the negative result of selective host depletion, can the authors comment or speculate on the idea that perhaps all of the bacteria in the skin mucus is already lysed (just extracellular DNA)?	We appreciate the reviewers suggestion. While it is possible that some bacterial cells in the skin mucus may be lysed prior to processing (which is discussed line 621-628), we consider it unlikely that all bacterial DNA would be extracellular. Fish skin is known to host a diverse microbial community adapted to this environment, many of which would require lysis for DNA extraction. Moreover, bacteria are generally resilient, and our host depletion methods were applied immediately after swabbing, thereby limiting the opportunity for degradation prior to processing.
4	Line 171: How were samples stored overnight if extractions were the next day? How did you ensure that DNases were not activated and degrading microbial DNA?	Thank you for raising this point. Samples were stored in TRIS-EDTA at –80 °C overnight to prevent DNA degradation, and we have clarified this in the manuscript (see lines 166–167 and 169).

5	Line 251: "All qPCR reactions"	We thank the reviewer for this comment. However, including the word reactions would make the phrase redundant ("quantitative polymerase chain reaction reactions"). For this reason, we have retained the original phrasing: "All qPCRs on samples...". (see line 241)
6	Line 434: Regarding the qPCR assay, I am concerned here about off target amplification and therefore a distorted signal: How do the authors know that the 18S target which was targeted and amplified from the fish was in fact from the fish? How do they know if the 16S target was actually 16S and not also amplifying 12S (mitochondria) which is very common? Their bacteria ratio might be inflated. I would strongly suggest running some of these qPCR reactions through shotgun metagenomics to verify the ratios of what is actually amplifying. This could have serious implications for what's happening. Alternatively, I would have considered using a more specific marker for the fish. This could be a marker which is very specific to the fish genome include a repeat region (microsat).	We appreciate the reviewer's concern regarding potential off-target amplification. To minimise this risk, primers were designed and validated through BLAST analysis against the NCBI non-redundant database, which confirmed that only trout 18S sequences matched our primer set. Furthermore, no amplification was observed in positive bacterial controls (Zymo mock community only), supporting the specificity of the 18S primers. While we acknowledge that other fish-specific markers (e.g., microsatellites) could be considered, we found the 18S gene to be sufficiently specific and appropriate for the aims of this study.
7	Figure 1: Could the authors display this another way: maybe a simple stacked bar of (% 16S, % 18S)? As of now it's a bit confusing as to what I'm looking at. You can keep the existing figure but maybe have a separate panel so its more intuitive.	We thank the reviewer for this suggestion. While we considered a stacked bar plot, we found it less suitable for our dataset: (i) it would require a log scale, making cross-sample comparisons difficult; (ii) over 160 individual bars would be needed, which would reduce clarity and make it unclear which elements should be compared; and (iii) such a figure would lack statistical support, and smaller visual differences could appear misleading.

8	Line 458: I am concerned about this conclusion of using mito ASVs as an endpoint. While 16S EMP primers do amplify 12S, its not the intended target. I think this is a reasonable thing to report but I don't believe you can make the argument that there are less mito overall and an inefficiency in host depletion. If authors really want to address this, they should design a specific mito DNA assay to amplify (e.g. qPCR). They could also do a rainbow trout specific marker. Check the RTqPCR literature and I'm sure there are primers out there for house keeping genes.	Thank you for this helpful comment. We agree that off-target amplification is not the most accurate measure of mitochondrial abundance. We have therefore retained the reporting of these data but reworded the statement to suggest this may be the case, rather than concluding that host depletion directly reduced mitochondrial DNA (see line 440). Given the consistency of the trend across 87 replicates, we still consider it noteworthy to report.
9	Line 488 Figure 4: These are some pretty highly variable numbers for alpha diversity. Were the statistical analyses used considering pairwise comparisons to control for the variation in alpha across biological samples? If not, that may be why there aren't statistical differences (biological variation is greater than kit variation). Please update	We confirm that the alpha diversity metrics presented are correct and that pairwise comparisons across sampling dates were accounted for using a generalised linear mixed model. Given the minimal differences observed in our qPCR results and beta diversity analyses, we think that it is not unexpected that little difference was detected in alpha diversity.
10	Line 496: Again, does this mean the same samples were processed in the control group along with all of the experimental groups? If not, the authors need to thoroughly describe how their samples used from the various kits may have differed and thus influenced the diversity metrics. For instance, were the fish used in the Molysis simply collected a different day or from a different tank? All of these minute factors can influence the fish microbiome.	Yes, all samples were processed identically. All fish were sourced from the same farm and maintained in the same tank. On each sampling date, we included as equal as possible numbers of fish for each host depletion method. Further details are provided in Table 1 (Line 410).
11	Line 523: Mycoplasma are likely intracellular here.	Thank you for this insight. We agree that Mycoplasma may be intracellular in this context, which could contribute to the observed depletion. However, in our manuscript we make no claims regarding their intra- or extracellular nature, and simply report their depletion in the CpG-methylated samples.

Reviewer #2

S/N	Comment	Response
1	Authors mentioned that data have been deposited in the European Nucleotide Archive (ENA) at EMBL-EBI under accession number PRJEB82663	Thank you

Bell et al. investigated different host DNA depletion techniques for profiling the skin microbiota of fish, using a freshwater species (rainbow trout) as a model organism. Several methods were applied to reduce host DNA contamination. This work is particularly interesting from a metagenomics perspective, as this approach stands to benefit significantly from effective host depletion. I found the study interesting and potentially useful, but it lacks critical information in several sections. My main concerns are 1) overgeneralization: The authors make general statements about applicability to "fish" based on results from a single species (juvenile rainbow trout). However, fish species may vary significantly in skin mucosa characteristics depending on habitat (freshwater vs saltwater), feeding strategies (herbivory vs carnivory), life stage (juveniles vs adults), and other ecological traits. These factors could influence how well host depletion methods work. The authors should clearly state that the study used a farmed rainbow trout and ensure the species name is included in the title, abstract, and throughout the manuscript. Although the primary focus is on methodology, the biological context is highly relevant and should not be overlooked. Some ecological background on the species used would be beneficial.	We thank the reviewer for this thoughtful comment. We agree that results from a single species should not be generalised to all fish without caution. To address this, we have clarified in the manuscript that our study used farmed juvenile rainbow trout (Oncorhynchus mykiss) as a representative model species (line 138). We also now highlight that ecological traits such as habitat (freshwater vs marine), diet, and life stage may influence host depletion outcomes and warrant further testing (line 692 to 700). At the same time, we emphasise that the methods evaluated here target broad host-cell features (e.g., CpG islands, eukaryotic cell structure, mucosal surfaces) that are widely conserved across fish taxa. For this reason, we expect our findings to be informative beyond rainbow trout, while recognising that additional studies in other species will be important for validation.
--	---

	2) Experimental design details: The manuscript lacks clarity on key aspects of the experimental setup: How many samples and replicates were used per fish? Were individual fish sampled multiple times (e.g., multiple swabs per fish), or were different fish used for each method? Were the swabs tested using multiple methods, or were separate swabs collected for each method? Individual variability between fish could influence the results, so it's important to report and acknowledge these details. 3) method selection rationale: were the five depletion methods selected based on prior success in other organisms? If so, which ones e.g. humans, marine animals, or other fish? Have these methods been tested before on mucosal layers from any aquatic animals?	We thank the reviewer for highlighting the need for greater clarity. We have now explicitly stated in the methods that each fish was sampled once, yielding a single independent sample per individual (lines 157-161). In total, 87 fish were sampled, and these 87 samples were distributed across the different host depletion methods (14–18 replicates per group). No fish were resampled. While individual variability between fish is expected, our design ensured balanced group sizes across time points and only tested for differences across methods (while accounting for different sampling days using a generalised linear mixed model), making the analysis robust to inter-individual variation. We also clarify that the five depletion methods were selected based on prior reports of effectiveness in mucosal or fish samples, as outlined in the Introduction (lines 104–115) and discussed further (lines 566–589).
--	---	---

	The section introducing the species and methodologies needs to be more clearly written. Consider revising the start of the methods section to clarify the sampling strategy and add a paragraph in the introduction with background on the 4 fish species.	We thank the reviewer for this comment. Lines 138–145 of the Methods section provide specific details on the study species (Oncorhynchus mykiss), including Latin name, weight, length, source, and husbandry. To improve clarity, we have also added wording to the sample collection paragraph (as outlined in our response to comment 3) to state explicitly that one sample was obtained per fish (n = 1). As the primary focus of this study is on host depletion methodologies rather than trout biology per se, we have kept the Introduction general to fish, microbiomes, and host depletion methods, while ensuring that the species used is clearly identified in the methods. We consider that adding ecological background on trout would not aid interpretation of our results and might risk distracting from the methodological focus of the study.
	5 Lines 76-78 and 96-100 are missing appropriate references.	We have added citations to support our statement on these lines (line 73; Wilson, 1997) (line 92; Oyola et al., 2013; Marotz et al., 2018; Heravi et al., 2020).
	6 Lines 147-168: It's unclear how many fish were sampled and why euthanasia was necessary if only skin mucosa was collected. Please clarify. Also, what are the sampling time points referenced, and why are they relevant?	We thank the reviewer for raising this point. We have clarified the sampling plan as outlined in our response to comment 3. The mention of euthanasia has been removed, as this was related to a separate experiment and is not relevant to the present study. (See line 153-154). To improve clarity, we now reference Table 1, which details the sampling time points (see line 410 for Table 1). These time points are relevant because samples collected on different dates must be statistically controlled for, as they could otherwise confound our results.

7	Line 177 suggests that all samples were extracted using the same CTAB/EDTA/chloroform method, but line 251 shows that different DNA extraction methods were used for each host depletion strategy. Please clarify which extraction protocol was used for each method and ensure consistency/clarity throughout the manuscript.	Line 251 refers to the different host depletion strategies and not DNA extraction, which we have better clarified (see line 241).
8	Line 427: Why do the sample numbers ("n=") in Table 1 differ between methods? Were these from the same fish sampled multiple times or from different fish? Clarify the sampling design.	We thank the reviewer for this comment. In Table 1, n refers to the number of samples at each time point (columns) and for each host depletion method (rows). Each n represents a new sample from a different fish, with no resampling performed (see line 410). As noted in our response to an earlier comment, we have clarified the sampling design in the manuscript to make this explicit (see response 3).
9	Line 438: Authors mentioned Figure 1, then Table 2. Figure 1 should be introduced before Table 2 if referenced earlier.	We thank the reviewer for noting this. As Table 2 appears before Figure 1 in our manuscript, we have revised the text so that the references now follow the correct order (see line 415 and 421).
10	Line 488: Differences shown in Figure 3 could be due to multiple factors. This goes back to my comment regarding the methods. It is essential to clarify the number of samples used and how they were collected to support the findings.	We thank the reviewer for this comment and agree that differences in alpha diversity could arise from multiple factors. In the figure caption, we simply state the alpha diversity metrics used without speculating on their differences. To support the findings, we have clarified in the methods that 87 fish were sampled once each (n = 1 per fish), with samples distributed across host depletion methods (14–18 replicates per group) (see response 3).
11	Line 637: Using "eDNA" may be confusing, as this typically refers to environmental DNA in the broader literature.	We thank the reviewer for this comment. In our manuscript, "extracellular DNA" was used in the context of DNA present outside intact cells. We recognise that "eDNA" is more commonly used in the broader literature to refer to environmental DNA, and to avoid confusion we have clarified our wording to ensure consistency with this usage (line 620).

	Lines 640-645: Is this speculation? References or clarification to support this would help.	We thank the reviewer for this helpful comment. We agree that the passage in question is speculative, and we have now clarified this by explicitly framing it as such. To support our interpretation, we have added references supporting our claim that that nuclease-based depletion methods, including MoLYsis, degrade not only host DNA but also extracellular microbial DNA, which may contribute to apparent depletion of certain taxa. We also cite studies demonstrating that such depletion methods can introduce bias into microbial community profiles (line 621 and 625).
	Lines 650-657: This section is difficult to follow. Please rephrase for clarity.	We thank the reviewer for highlighting that this section was difficult to follow. We have revised the text to improve clarity while retaining the original meaning and supporting references (see lines 631 to 640).
	Lines 687-692: DNA from mucosa can be challenging to extract. Was the DNA integrity assessed (e.g., using a 0.8-1% agarose gel to check for high molecular weight DNA)? DNA fragmentation could explain low bacterial DNA recovery in MAGs.	We thank the reviewer for this helpful comment. As part of the Exeter Sequencing Centre's quality control process, all submitted DNA was assessed for high molecular weight using either agarose gel electrophoresis or an Agilent D1000 screentape system. Only DNA that passed this quality check was accepted for sequencing. While the results of these checks were not recorded for our samples (as they were performed by the sequencing centre and inspected visually), all DNA used in this study therefore met the required integrity standards. We have added this to our methods for additional clarity (see lines 381-384).

Re: Spectrum01838-25R1 (Assessment of the Effectiveness of Host Depletion Techniques for Profiling Fish Skin Microbiomes and Metagenomic Analysis)

Dear Dr. Ashley G Bell:

Your manuscript has been accepted, and I am forwarding it to the ASM production staff for publication. Your paper will first be checked to make sure all elements meet the technical requirements. ASM staff will contact you if anything needs to be revised before copyediting and production can begin. Otherwise, you will be notified when your proofs are ready to be viewed.

Sincerely,
Justine Debelius
Editor
Microbiology Spectrum

Reviewer #2 (Comments for the Author):

I appreciate the authors for addressing my comments thoroughly. I am satisfied with the revisions and have no further concerns.